# Changes in Dietary Nutrient Intake and Estimated Glomerular Filtration Rate over a 5-Year Period in Renal Transplant Recipients

**DOI:** 10.3390/nu16010148

**Published:** 2023-12-31

**Authors:** I-Hsin Lin, Yi-Chun Chen, Tuyen Van Duong, Shih-Wei Nien, I-Hsin Tseng, Yi-Ming Wu, Hsu-Han Wang, Yang-Jen Chiang, Chia-Yu Chiang, Chia-Hui Chiu, Ming-Hsu Wang, Nien-Chieh Yang, Te-Chih Wong

**Affiliations:** 1Department of Medical Nutrition Therapy, Linkou Chang Gung Memorial Hospital, Taoyuan 333, Taiwan; cabbage@cgmh.org.tw (I.-H.L.); nina0904@cgmh.org.tw (S.-W.N.); cathy40422@cgmh.org.tw (I.-H.T.); yiming1023@cgmh.org.tw (Y.-M.W.); 2School of Nutrition and Health Sciences, College of Nutrition, Taipei Medical University, Taipei 110, Taiwan; yichun@tmu.edu.tw (Y.-C.C.); tvduong@tmu.edu.tw (T.V.D.); 3Department of Urology, Linkou Chang Gung Memorial Hospital, Taoyuan 333, Taiwan; seanwang@cgmh.org.tw (H.-H.W.); zorro@cgmh.org.tw (Y.-J.C.); 4Department of Medicine, Chang Gung University, Taoyuan 333, Taiwan; 5Department of Business Administration, College of Management, National Changhua University of Education, Changhua 500, Taiwan; cychiang@cc.ncue.edu.tw; 6Center for General Education, Taipei Medical University, Taipei 110, Taiwan; thera@tmu.edu.tw (C.-H.C.); mattwang@tmu.edu.tw (M.-H.W.); 7Department of Nutrition and Health Sciences, Chinese Culture University, Taipei 111, Taiwan; a9221185@ulive.pccu.edu.tw

**Keywords:** renal transplant recipients, dietary nutrients, renal function, glomerular filtration rate, dietary reference intakes, Dietitian Association Australia

## Abstract

The scarcity of dietary guidance for renal transplant recipients (RTRs) raises concerns regarding obesity and associated comorbidities, including impaired renal function. This two-stage cross-sectional study examined longitudinal changes in dietary nutrient intake in the same individuals over a 5-year interval. This study involved two stages: T1 (September 2016 to June 2018) and T2 (July 2022 to August 2023). The average duration between the two data collection stages was 6.17 ± 0.42 (range 5.20–6.87) years. The study included 227 RTRs with an average age and time since transplant of 49.97 ± 12.39 and 9.22 ± 7.91 years, respectively. Of the 35 patients who participated in both phases, fewer than half met the recommended intakes for energy, dietary fiber, and most vitamins and minerals, as set in the Dietary Reference Intakes (DRIs) or by the Dietitian Association Australia (DAA). Over half exceeded the DRI recommended intake for total protein, and more than 80% of the protein consumed per kilogram of body weight exceeded the DAA’s recommendations. In the T2 stage, the RTRs had a significantly higher blood urea nitrogen level, lower albumin level, and estimated glomerular filtration rate. These findings indicate that deteriorating dietary intake in RTRs can adversely affect their nutritional status and transplanted kidney function over a 5-year period.

## 1. Introduction

A renal transplant is a crucial treatment for patients with end-stage renal disease. Although a renal transplant effectively prolongs survival [1] and improves quality of life [2], it concurrently increases the incidence of metabolic complications in renal transplant recipients (RTRs) [3]. Obesity is a prevalent comorbidity in RTRs and adversely affects the transplanted kidney’s function and a patient’s long-term survival [4]. Furthermore, obesity increases the risks of insulin resistance, diabetes, hypertension, and dyslipidemia in RTRs, thereby increasing the risk of cardiovascular diseases (CVDs) [5,6].

Nearly 50% of RTRs exhibit postoperative weight gain, regardless of their preoperative obesity status [7,8]; this gain is an average increase of 10–30% of their initial body weight [7,9,10,11]. According to Chan et al. [12], weight gain in RTRs is attributable to various factors, such as the use of immunosuppressants, the perception of improved health due to enhanced renal function, alleviation of uremia, or a sense of liberation from pretransplant dietary restrictions. These factors, in turn, lead to improper dietary intake, resulting in high risks of obesity and metabolic complications. Therefore, given the global challenge of limited availability of organs for transplant, ongoing research focused on nutritional care management for RTRs is crucial. Furthermore, effective management necessitates a thorough examination of factors contributing to the postrenal transplant occurrences of obesity, dyslipidemia, and other cardiovascular risk factors, with particular emphasis on dietary factors.

Nutritional adequacy refers to the sufficient intake of dietary nutrients to fulfill nutritional needs, ensuring not only optimal health but also the prevention of chronic illnesses. The lack of relevant literature makes it difficult to establish specific dietary guidelines tailored for RTRs in Taiwan. In addition, few comprehensive long-term studies examining fluctuations in dietary nutrient intake in RTRs have been conducted in Taiwan and globally. To address this gap, the present study employed an observational approach to assess longitudinal changes in the nutrient intake of individuals over a period of at least 5 years. We compared the findings with the Dietary Reference Intakes (DRIs) applicable to the Taiwanese population [13] and the reference nutrient intake recommendations provided by the Dietitian Association Australia (DAA) for RTRs. The DAA is one of the few international sources of guidance on long-term nutrient intake and the management of chronic complications in this population [14]. We hypothesized that specific deteriorations in dietary intake are expected because an unrestricted and liberalized diet may lead to nutrition problems within this population.

## 2. Participants and Methods

### 2.1. Study Design

This study was approved by the Institutional Review Board of the Chang Gung Medical Foundation (IRB number: 201600954B0 and 202201338B0C601) and followed a two-stage cross-sectional design. The initial phase, conducted from September 2016 to June 2018 (referred to as time point 1, T1 Stage), was followed by the second phase, spanning from July 2022 to August 2023 (referred to as time point 2, T2 stage). Eligible patients were recruited from the outpatient clinic of the Urology and Renal Transplantation Department. Upon obtaining their consent, patients’ basic information, dietary details, anthropometric measurements, and laboratory test results were collected by well-trained dietitians and staff during regular follow-up visits, as described previously [15].

### 2.2. Patient Recruitment

This study recruited outpatients from Linkou Chang Gung Memorial Hospital via advertisements. To be eligible for inclusion, patients had to have undergone renal transplant at least 6 months prior and be on immunosuppressive therapy, which included a regimen of calcineurin inhibitors, antimetabolites, and steroids before recruitment. We included patients who were aged 20 years or above, consumed food orally, and exhibited no signs of acute rejection, substantial changes (>25%) in the glomerular filtration rate (GFR), or infection confirmed using physical examination within the 3 months prior to recruitment. Patients with a history of more than two renal transplants or other organ transplants and extreme values in dietary intake (<800 or >3000 kcal) were excluded.

In the T1 stage, 106 patients were initially evaluated for eligibility, resulting in the enrolment of 90 RTRs. Subsequently, five patients were excluded —four due to extreme energy intake and one due to incomplete data collection. In the T2 stage, 142 individuals participated in data collection. A comparative analysis of the baseline information of patients participating in both stages is presented in Appendix A. Our analyses included data from 35 RTRs who participated in both phases.

### 2.3. Demographics and Anthropometric Data

We collected the patients’ demographic information—including sex, age, kidney transplant date, source of the transplanted kidney, and blood pressure—via medical record reviews and cross-checked this information during their follow-up visits to ensure accuracy. Anthropometric data, including height (measured without shoes) and weight (measured after fasting, without shoes, and in light clothing), were assessed during the patients’ follow-up visits under fasting conditions to enable the calculation of the patient’s body mass index (BMI).

### 2.4. Laboratory Tests

Blood samples, collected concurrently with the patients’ dietary records within the same month, were obtained following at least 8 h of fasting. These samples were analyzed at Chang Gung Memorial Hospital’s clinical laboratories using an automated analyzer (Sysmex XN-3000, Kobe, Japan) and standardized methodologies. The investigated parameters included albumin, blood urea nitrogen, creatinine, total cholesterol, triglycerides, high-density lipoprotein cholesterol, glycated hemoglobin A1C, insulin, uric acid, and high-sensitivity C-reactive protein. Insulin resistance was determined using the homeostatic model assessment of insulin resistance with the following formula: insulin (U/mL) × blood glucose concentration (mmol/L) ÷ 22.5 [16]. The estimated GFR (eGFR), retrieved from patients’ medical records, had been derived using the Modification of Diet in Renal Disease equation [17]:eGFR (mL/min/1.73 m^2^) = 175 × (serum creatinine) − 1.154 × (age) − 0.203 × 0.742 (if female) × 1.21 (if African American)

### 2.5. Dietary Data

The patients were required to complete dietary records covering the 3 days preceding their follow-up visit (two weekdays and one weekend day). The record book provided straightforward estimations of food portions (such as spoons, bowls, and cups). On the day of the patients’ follow-up visit, the content and portion estimations in the diet record were confirmed and validated via a face-to-face interview with a dietitian. We used the COFIT Pro online software (version 1.0.0; Cofit HealthCare, Taipei, Taiwan) [18], which relies on the Taiwan Food Composition Table provided by the Ministry of Health and Welfare (MOHW) [19], to analyze the participants’ dietary nutrient intake. We compared their intake with the latest DRIs proposed by Taiwan’s MOHW [13] and the DAA [14]. Taiwan’s DRIs provide recommendations based on sex and age. We considered nutrient intake to align with recommendations when it reached at least 70% of the DRIs’ suggested values [13].

### 2.6. Statistic Analysis

Statistical analyses were performed using SAS software, version 9.4 (SAS Institute, Cary, NC, USA). Statistical significance was determined at a *p* value of < 0.05. For all cases, variable distributions were assessed using the Shapiro–Wilk test and Q–Q plot. Quantitative variables are presented as the mean ± SD or as a number and percentage. The paired t test or Wilcoxon signed-rank test was used to compare variables in 35 RTRs at the T1 and T2 stages.

## 3. Results

The baseline characteristics of the 227 RTRs recruited in two stages are summarized in Appendix A. The participants had a mean age of 49.97 ± 12.39 years, and upon their enrolment, it had been 9.22 ± 7.91 years since their transplant operation. Of the included RTRs, 68.7% (*n* = 156) received their transplant from a deceased donor. The average energy intake of the RTRs was 1793.61 ± 457.02 kcal. The T2 stage participants exhibited significantly lower levels of energy intake (1746.34 ± 493.70 vs. 1872.58 ± 377.8 kcal) and carbohydrate intake (182.02 ± 59.09 vs. 207.22 ± 47.34 g) and a significantly lower percentage of carbohydrates in their total energy intake (41.79% ± 8.38% vs. 44.53% ± 6.46%) than did the T1 stage participants. In addition, the T2 stage participants exhibited significantly higher percentages of protein (16.21% ± 2.91% vs. 14.46 ± 1.76%) and fats (42.41% ± 7.17% vs. 40.55% ± 5.71%) in their total energy intake than did the T1 stage participants. Of the six primary categories of foods, the T2 stage participants had a significantly lower intake of whole grains and cereals (8.89 ± 3.30 vs. 10.56 ± 2.69 servings) than did the T1 stage participants. Moreover, we noted significantly greater consumption of soybeans, fish, eggs, and meat (6.58 ± 2.48 vs. 5.78 ± 1.62 servings) in the T2 stage participants than in the T1 stage participants.

Table 1 displays the longitudinal changes in characteristics observed in the 35 RTRs involved in both the T1 and T2 stages. The average time interval between the two data collection points was 6.17 ± 0.42 years, ranging from a minimum of 5.20 years to a maximum of 6.87 years. The average weights in the T1 and T2 stages were 65.42 ± 13.29 and 65.07 ± 13.85 kg, respectively, and the average BMI values were 24.46 ± 3.84 and 24.52 ± 3.80 kg/m², respectively. We discovered no significant change in the participants’ weight or BMI between the two time points. Regarding laboratory findings, the T2 stage participants had significantly higher levels of blood urea nitrogen (26.71 ± 9.79 vs. 22.63 ± 7.55 mg/dL; *p* = 0.022) and significantly lower levels of albumin (4.14 ± 0.28 vs. 4.34 ± 0.30 g/dL; *p* = 0.0001) than did the T1 stage participants. Moreover, the eGFR was significantly lower (51.09 ± 16.89 vs. 55.31 ± 14.05 mL/min/1.73 m²; *p* = 0.044) in the T2 stage participants than in the T1 stage participants.

Regarding energy and macronutrient intake, the participants exhibited significantly lower carbohydrate intake (184.40 ± 53.31 vs. 208.17 ± 45.66 g) at the T2 stage than at the T1 stage. In addition, we found a notable increase in the percentage of protein (15.65% ± 1.85% vs. 14.39% ± 1.72%), saturated fatty acids (10.98% ± 2.20% vs. 9.25% ± 2.71%), and monounsaturated fatty acids (30.97 ± 11.13 vs. 26.41 ± 8.52 g) in the total energy intake and their percentage of the total energy intake (15.80% ± 4.39% vs. 12.57% ± 3.28%).

Table 2 lists the number and percentage of the RTRs whose dietary nutrient intake aligned with the recommendations of the DRIs or DAA. At T1, 7 out of the 35 participants (20.0%) met the recommended energy intake set in the DRIs; this number increased to 15 (42.9%) at T2. Approximately 60% of the RTRs met the DRIs’ recommended protein intake. Additionally, 29 RTRs (82.9%) met the DAA recommendations for protein intake per kilogram of body weight in both stages. However, only one RTR met the DRIs’ recommendations for total fat as a percentage of energy intake.

Further analysis revealed that 21 RTRs (60.0%) met the DRIs’ recommendations for saturated fatty acid intake at T1; this number decreased to 10 (28.6%) at T2. Conversely, at T1, 35 RTRs (100%) and 3 RTRs (8.6%) met the recommended percentages of monounsaturated fatty acids and n-6 polyunsaturated fatty acids in total energy intake, respectively. At T2, 28 RTRs (80.0%) and 3 RTRs (8.6%) met the respective DAA recommendations for these fatty acids. Few RTRs met the DRI recommendations for dietary fiber intake, with only 1 (2.9%) at T1 and 3 (8.6%) at T2.

At T1, more than half of the RTRs met the DRI recommendations for specific vitamins, including vitamin A (24 participants, 68.6%) and vitamin B12 (18 participants, 51.4%). At T2, this trend was observed for vitamin B1 (20 participants, 57.1%), niacin (18 participants, 51.4%), and vitamin B12 (19 participants, 54.3%). However, for vitamin B1 at T1, vitamins E, C, B2, and B6, and folic acid at T2, more than half of the RTRs did not meet the DRI recommendations. Regarding mineral intake, less than half of the RTRs met the DRI recommendations for calcium, magnesium, iron, and zinc intake in both the T1 and T2 stages.

Table 3 presents the number and percentage of the participants who achieved more than 70% of the DRI recommended intake of vitamins and minerals in the two stages. Although more than half of the participants met the DRI recommended intake of greater than 70% for most nutrients, fewer than half met the requirements for folic acid, magnesium, and zinc. In addition, the number of the RTRs meeting the targets for vitamins A, C, and B12 and for iron was lower at the T2 stage, indicating a notable decline in the meeting of these specific nutrient goals over the 5-year interval.

## 4. Discussion

In this study, the participants exhibited no significant change in weight or BMI between the T1 and T2 stages. However, concerning laboratory data, we noted significant deteriorations in indicators related to nutritional status (e.g., albumin) and transplant kidney function (e.g., blood urea nitrogen and eGFR). These findings indicate that RTRs’ inappropriate postoperative lifestyle choices, including poor dietary patterns, adversely affect their nutritional status and the function of their transplanted kidney. Dietary nutrient intake analysis revealed that less than half of the participants in both stages met the DRI or DAA’s recommended intake for overall dietary energy, dietary fiber, most vitamins, and minerals. Moreover, over half of the participants exceeded the DRI recommended intake for total protein, and more than 80% of the participants consumed a higher percentage of protein per kilogram of body weight than that recommended by the DAA. Such dietary imbalances may increase the risk of metabolic abnormalities.

Recommendations for RTRs’ protein intake should be tailored on the basis of the individual’s renal function. In this study, we observed that the eGFR values of the RTRs over a 5-year interval ranged from 55.31 ± 14.05 to 51.09 ± 16.89 mL/min/1.73 m^2^, which corresponds to chronic kidney disease (CKD) stage 3a [20]. Patients in this category are recommended to consume less protein compared with healthy adults. Bernardi et al. determined that maintaining an appropriate protein intake (0.8 g/kg) over an extended period led to unchanged renal function after 12 years in RTRs. By contrast, patients with higher protein intake (1.4 g/kg) experienced a 40% decline in renal function [21,22]. At both the T1 and T2 stages, the protein intake of our participants was approximately 1.1 g/kg, and more than 80% of the RTRs exceeded the recommendations by the DAA, potentially increasing the workload on the transplanted kidney. Such excessive protein intake may trigger renal arteriolar vasodilation, leading to glomerular hyperfiltration and subsequent glomerular damage [23,24].

In this study, only 2.9% of the RTRs adhered to the recommended percentages of total fat in their total energy intake. In addition, the overall intake percentages of both total fat and saturated fatty acids for the RTRs involved in both stages did not align with the DRI recommended guidelines of 20–30% and less than 10% of total energy intake, respectively. Sun et al. observed that, in experimental mice, a high-fat diet-induced damage to glomerular and renal tubular structures, increased oxidative stress, and triggered apoptosis in renal tubular cells via mitochondrial fission [25]. In addition, Ruiz-Núñez et al. highlighted that excessive saturated fatty acid intake might cause mild systemic inflammation and alter lipoprotein metabolism, elevating the risk of CVD [26]. Furthermore, only one patient in the present study met the DRI recommendation for dietary fiber. Emerging evidence highlights the numerous health benefits associated with high fiber intake in relation to the occurrence, progression, and complications of CKD [27]. Xu et al. found that older men in Swedish communities with higher dietary fiber intake exhibited better kidney function [28]. Moreover, several studies have suggested that dietary fiber exerts anti-inflammatory effects, potentially offering protection against CVD in RTRs and healthy individuals [29,30].

Evidence directly linking carbohydrate intake to transplanted kidney function remains limited. In this study, the participants were found to consume significantly less carbohydrates in the second phase compared with the initial phase, with the average percentage of total carbohydrates in the diet falling below 45% at the T2 stage. Using data from the National Health and Nutrition Examination Survey 1999–2018, Dustin et al. revealed that carbohydrate intake below the recommended level (45% of energy) in conjunction with a high-fat diet was associated with a higher prevalence of metabolic syndrome in adults in the United States [31]. Given that metabolic syndrome is a critical and prevalent risk factor for RTRs [32], establishing acceptable macronutrient distribution ranges that can help prevent metabolic complications in this population is essential.

In this study, less than 40% of the RTRs met over 70% of the DRI recommended intakes for folic acid, calcium, magnesium, and zinc in their diet. Furthermore, the number of RTRs meeting the 70% DRI recommendation for vitamins A, C, and B12 and iron decreased within the 5-year period. This trend might be associated with the RTRs continuing a dietary pattern from their dialysis days [33], resulting in low consumption of dairy products, dark green vegetables, nuts, and other foods rich in these vitamins and minerals. Cianciolo et al. suggested that the insufficient intake of folic acid and vitamin B12 in patients with CKD resulted in an elevated blood homocysteine level, potentially increasing the risks of CVD and CKD progression [34]. In addition, Takahashi et al. reported that a low blood vitamin C level in patients with CKD but without diabetes might elevate oxidative stress, leading to vascular endothelial cell dysfunction and worsening kidney function [35]. A previous study detected iron deficiencies in 62.4% of long-term renal recipients without anemia [36], and cyclosporine-treated RTRs exhibited a significantly decreased serum magnesium level [37]. Finally, zinc deficiency was identified as a risk factor for CKD progression [38]. To summarize, RTRs’ inadequate dietary nutrient intake can adversely affect their prognosis.

To the best of our knowledge, this study is the first to compare dietary nutrient intakes in the same group of RTRs over a minimum of 5 years, particularly within the Asian context. However, this study has some limitations that should be considered. First, because of the cross-sectional study design, causality between changes in dietary nutrient intake and the nutritional status or transplant kidney function of patients could not be inferred. Nevertheless, via repeated cross-sectional studies, we determined dynamic relationships between nutrient intake and the eGFR by comparing nutrient intake and adherence to DRI and DAA recommendations. Additional comprehensive prospective studies and controlled trials should be conducted to validate our research findings. Second, this study is limited by a small sample size, lacking the ability to represent the entire RTR population. To the best of our knowledge, this is the first study to address within-subject longitudinal changes in nutrient intake over an average period of 6 years. For a considerable period, our research team has been dedicated to studying the effects of RTRs’ diet on their nutritional status, transplanted kidneys, and postoperative prognosis. Moving forward, we will persistently monitor data from current patients to better comprehend the interconnectedness and effect of their dietary habits on their overall health status and outcomes. Third, due to constraints related to research funding, this study did not assess blood test data corresponding to inadequate dietary nutrient intake. Furthermore, in the analysis of dietary vitamin D deficiency, data from the Taiwan Food Composition Table for this was lacking, and this aspect was consequently omitted from this investigation. Fifth, this study investigated only the association between nutrition and health, relying on dietary nutrient intake. Future studies should employ diet quality assessment indicators that encompass both food and nutrient intake to explore the association between adherence to dietary recommendations and the prevalence of chronic diseases. Sixth, we estimated the daily total energy requirements of patients using the simplified calorie algorithm, taking into account only the body mass index of the subjects and their daily activity levels. Given that indirect calorimetry is more accurate, caution is advised when providing nutritional counseling to patients based on formulas that often fail to accurately determine the correct intake. This could potentially lead patients to consume either excessive or insufficient calories. Finally, future studies should address various unknown confounding factors, such as the metabolic effects of immunosuppressive medications, the specific primary kidney condition that resulted in end-stage kidney disease, and factors including the body composition and exercise routines of RTRs.

## 5. Conclusions

We observed marked shifts in the dietary nutrient intake of RTRs over a period of at least 5 years; for most nutrients, the RTRs’ intakes did not meet the recommendations set in the DRIs and by the DAA. In particular, deficiencies were noted in carbohydrate, vitamin, and mineral intake, whereas fat and protein intake exceeded the recommended amounts. Moreover, the indicators of renal function (eGFR) and nutritional status (albumin) demonstrated a significant downward trend over time. Thus, RTRs should be regularly assessed by a dietitian, and their nutrition-related indicators should be monitored.

On the basis of the findings of this study, we recommend that dietitians and professionals in the field of nutrition focus on advocating a balanced diet when delivering dietary education strategies to RTRs in the future. This approach should involve providing individualized recommendations on portion sizes for macronutrients and the six primary food categories. Adherence to an energy intake range should be encouraged, and patients should diversify their food choices by opting for various nutrient-dense foods that align with recommended intake levels. This strategy aims to mitigate the risk of inadequate vitamin and mineral intake while concurrently reducing the potential deterioration of nutritional status and decline in the functionality of the transplanted kidney.

## Figures and Tables

**Table 1 nutrients-16-00148-t001:** Baseline demographic, anthropometric, clinical, and nutritional data of the same RTRs over a 5-year interval (n = 35) ^1,2^.

	T1 Stage	T2 Stage			T1 Stage	T2 Stage	
Numbers	35	35	*p* Value		35	35	*p* Value
**Demographics**				**Dietary intake**			
Male/female	21/14	21/14		Energy, kcal/day	1898.96 ± 376.87	1753.53 ± 446.24	0.178
Age, year	49.31 ± 9.72	55.06 ± 9.77	<0.0001	Carbohydrate, g/day	208.17 ± 45.66	184.40 ± 53.31	0.017
Renal transplant time, year	10.17 ± 5.43	16.72 ± 5.76	<0.0001	Carbohydrate, % energy	44.08 ± 5.82	42.22 ± 6.43	0.071
Tacrolimus/cyclosporine used	23/12	15/14 ^$^		Protein, g/day	67.9 ± 13.53	68.20 ± 16.90	0.761
Deceased/living donors	29/6	29/6		Protein, g/kg/day	1.08 ± 0.32	1.10 ± 0.37	0.742
**Anthropometry**				Protein, % energy	14.39 ± 1.72	15.65 ± 1.85	0.002
Height, cm	163.07 ± 7.70	162.27 ± 7.66	0.072	Fat, g/day	86.23 ± 22.93	82.69 ± 25.00	0.688
Body weight, kg	65.42 ± 13.29	65.07 ± 13.85	0.879	Fat, % energy	40.51 ± 5.65	42.20 ± 6.15	0.157
Body mass index, kg/m^2^	24.46 ± 3.84	24.52 ± 3.80	0.736	SFA, g/day	19.60 ± 7.13	21.82 ± 8.47	0.175
**Laboratory**				SFA, % energy	9.25 ± 2.71	10.98 ± 2.20	0.002
Albumin, g/dL	4.34 ± 0.30	4.14 ± 0.28	0.0001	MUFA, g/day	26.41 ± 8.52	30.97 ± 11.13	0.032
Blood urea nitrogen, mg/dL	22.63 ± 7.55	26.71 ± 9.79	0.022	MUFA, % energy	12.57 ± 3.28	15.80 ± 4.39	0.000
Creatinine, mg/dL	1.31 ± 0.34	1.42 ± 0.43	0.156	PUFA, g/day	28.75 ± 12.76	29.23 ± 13.10	0.974
Total cholesterol, mg/dL	206.69 ± 39.28	179.14 ± 37.73	0.002	PUFA, % energy	13.50 ± 5.12	15.07 ± 5.65	0.157
Triglycerides, mg/dL	141.74 ± 98.49	133.57 ± 73.72	0.804	Cholesterol, mg/day	239.69 ± 135.13	265.64 ± 106.24	0.325
HDL-C, mg/dL	54.09 ± 16.79	52.46 ± 17.14	0.527	Fiber, g/day	12.83 ± 6.37	11.83 ± 5.70	0.384
HbA1c, %	5.77 ± 1.15	6.02 ± 0.73	0.239	Na, mg/day	1143.05 ± 931.31	1026.71 ± 858.87	0.520
Insulin, U/mL	10.74 ± 19.82	7.80 ± 6.79	0.204	Ca, mg/day	314.99 ± 124.06	488.16 ± 418.75	0.008
Uric acid, mg/dL	6.14 ± 1.36	5.77 ± 1.27	0.345	Mg, mg/day	177.82 ± 59.03	220.97 ± 124.44	0.015
hs-CRP, mg/dL	6.05 ± 17.85	4.25 ± 8.31	0.618	P, mg/day	713.91 ± 211.56	854.04 ± 233.61	0.011
**Six primary food categories**				K, mg/day	1741.78 ± 570.36	2030.39 ± 803.50	0.061
Whole grains and cereals, servings/day	10.53 ± 2.67	9.69 ± 3.11	0.189	Iron, mg	8.78 ± 2.41	9.94 ± 8.53	0.476
Soybeans, fish, eggs, and meat, servings/day	6.09 ± 1.65	6.47 ± 1.96	0.375	Zinc, mg	8.83 ± 2.34	8.78 ± 2.64	0.879
Dairy products, servings/day	0.13 ± 0.24	0.23 ± 0.46	0.319	Vitamin B1, mg/day	1.00 ± 0.28	1.14 ± 0.39	0.123
Vegetables, servings/day	2.46 ± 1.04	2.43 ± 1.18	0.879	Vitamin B2, mg/day	0.80 ± 0.17	0.95 ± 0.39	0.032
Fruits, servings/day	1.19 ± 1.00	0.62 ± 0.90	0.008	Niacin, mg/day	11.60 ± 3.31	15.35 ± 4.25	0.000
Oils, fats, nuts and seeds, servings/day	9.79 ± 3.31	10.05 ± 3.59	0.653	Vitamin B6, mg/day	1.22 ± 0.33	1.50 ± 0.51	0.007
**Others**				Vitamin B12, μg/day	2.64 ± 1.31	3.16 ± 2.27	0.279
eGFR, mL/min/1.73 m^2^	55.31 ± 14.05	51.09 ± 16.89	0.044	Folic acid, μg/day	182.55 ± 74.73	245.31 ± 110.73	0.005
SBP, mmHg	137.22 ± 17.73	136.19 ± 16.77	0.745	Vitamin C, mg/day	100.29 ± 63.78	114.18 ± 98.14	0.936
DBP, mmHg	79.29 ± 12.99	76.81 ± 11.28	0.353	Vitamin A, μg RE/day	689.02 ± 299.87	622.02 ± 390.35	0.469
HOMA-IR	3.00 ± 7.36	1.92 ± 1.82	0.374	Vitamin E, mg α-TE/day	9.35 ± 3.22	16.88 ± 28.60	0.134

Abbreviations: RTRs, renal transplant recipients; HDL-C, high-density lipoprotein cholesterol; HbAlC, glycated hemoglobin A1c; hs-CRP, high-sensitivity C-reactive protein; eGFR, estimated glomerular filtration rate; SBP, systolic blood pressure; DBP, diastolic blood pressure; HOMA-IR, homeostasis model assessment–estimated insulin resistance; SFA, saturated fatty acid; MUFA, monounsaturated fatty acid; PUFA, polyunsaturated fatty acid, Na, sodium; Ca, calcium; Mg, magnesium; P, phosphorous; K, potassium; RE, retinol equivalent; TE, tocopherol equivalent. ^1^ This two-stage cross-sectional study was conducted from September 2016 to June 2018, referred to as the T1 stage, followed by the T2 stage, spanning from July 2022 to August 2023. Data are presented as the mean ± standard deviation or number, as appropriate. ^2^ Statistical analyses were conducted using the paired *t*-test or Wilcoxon sign rank test, as appropriate. ^$^ No records of six patients.

**Table 2 nutrients-16-00148-t002:** Number and percentage of RTRs whose dietary nutrient intake aligned with the recommendations of the DRIs or DAA (n = 35) ^1^.

Items	Daily Intake		Daily Recommendations	
	T1 Stage n (%)	T2 Stage n (%)	DRIs	DAA
Energy, kcal	7 (20.0)	15 (42.9)	M: 1650–2650 kcal; F: 1300–2100 kcal	
Protein, g	20 (57.1)	22 (62.9)	M: 70 g; F: 60 g	
Protein, g/kg BW	29 (82.9)	29 (82.9)		M: 0.84 g/kg; F: 0.75 g/kg
Fat, % of energy	1 (2.9)	1 (2.9)	20–30% of energy	
SFA, % of energy	21 (60.0)	10 (28.6)	<10% of energy	
MUFA, % of energy	35 (100.0)	28 (80.0)		≤20% of energy
n-6 PUFA, % of energy	3 (8.6)	3 (8.6)		8–10% of energy
Fiber, g	1 (2.9)	3 (8.6)	M: 23–37 g; F: 18–29 g	
Vitamins				
Vitamin A, μg RE	24 (68.6)	14 (40.0)	M: 600 μg RE ; F: 500 μg RE	
Vitamin E, mg α-TE	5 (14.3)	14 (40.0)	12 mg α-TE	
Vitamin C, mg	14 (40.0)	11 (31.4)	100 mg	
Vitamin B1, mg	11 (31.4)	20 (57.1)	M: 1.2 mg; F: 0.9 mg	
Vitamin B2, mg	2 (5.7)	7 (20.0)	M: 1.3 mg; F: 1.0 mg	
Niacin, mg	6 (17.1)	18 (51.4)	M: 16 mg; F: 14 mg	
Vitamin B6, mg	5 (14.3)	12 (34.3)	1.5–1.6 mg	
Vitamin B12, μg	18 (51.4)	19 (54.3)	2.4 μg	
Folic acid, μg	1 (2.9)	2 (5.7)	400 μg	
Minerals				
Calcium, mg	0 (0.0)	1 (2.9)	1000 mg	
Magnesium, mg	1 (2.9)	4 (11.4)	M: 350–380 mg; F: 300–320 mg	
Phosphorus, mg	10 (28.6)	22 (62.9)	800 mg	
Iron, mg	7 (20.0)	11 (31.4)	M: 10 mg; F: 10–15 mg	
Zinc, mg	2 (5.7)	2 (5.7)	M: 15 mg; F: 12 mg	

Abbreviations: RTRs, renal transplant recipients; DRIs, dietary reference intakes; DAA, Dietitian Association of Australia; M, men; F, women; BW, body weight; SFA, saturated fatty acid; MUFA, monounsaturated fatty acid; PUFA, polyunsaturated fatty acid; RE, retinol equivalent; TE, tocopherol equivalent. ^1^ Data are expressed as the number and percentage. The T1 stage was conducted from September 2016 to June 2018, followed by the T2 stage, spanning from July 2022 to August 2023.

**Table 3 nutrients-16-00148-t003:** Number and percentage of RTRs achieving more than 70% of the DRI recommended intake of vitamins and minerals in the two stages ^1^.

Items	70% DRIs Recommendations	
	T1 Stage, n (%)	T2 Stage, n (%)
Vitamins		
Vitamin A, μg RE	30 (85.7)	24 (68.6)
Vitamin E, mg α-TE	19 (54.3)	26 (74.3)
Vitamin C, mg	23 (65.7)	18 (51.4)
Vitamin B1, mg	31 (88.6)	31 (88.6)
Vitamin B2, mg	13 (37.1)	21 (60.0)
Niacin, mg	19 (54.3)	30 (85.7)
Vitamin B6, mg	20 (57.1)	28 (80.0)
Vitamin B12, μg	27 (77.1)	26 (74.3)
Folic acid, μg	1 (2.9)	13 (37.1)
Minerals		
Calcium, mg	0 (0.0)	6 (17.1)
Magnesium, mg	5 (14.3)	10 (28.6)
Phosphorus, mg	25 (71.4)	31 (88.6)
Iron, mg	28 (80.0)	26 (74.3)
Zinc, mg	12 (34.3)	13 (37.1)

Abbreviations: RTRs, renal transplant recipients; DRIs, dietary reference intakes; RE, retinol equivalent; TE, tocopherol equivalent. ^1^ Data are expressed as the number and percentage. The T1 stage was conducted from September 2016 to June 2018, followed by the T2 stage, spanning from July 2022 to August 2023.

## Data Availability

The data presented in this study can be obtained by contacting the corresponding author upon request. Ethical restrictions prevent public availability of the data.

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
