# Peer review of "Changes in Dietary Nutrient Intake and Estimated Glomerular Filtration Rate over a 5-Year Period in Renal Transplant Recipients"

_nutrients, 2023, doi:10.3390/nu16010148_

Round 1
Reviewer 1 Report
Comments and Suggestions for Authors
This great study shows that a good nutritional status is mandatory to preserve kidney transplants, by examining the evolution over a sufficiently long time, in a well underbuild setup with good statistics. I have some remarks:
Methods: Was the advised caloric intake of patients calculated or measured, when providing initial advice to patients? I assume the former?
Discussion: In the absence of indirect calorimetry, nutritional counseling could have caused patients to take either too much or too few calories (due to formulas often missing the actual correct intake), which should be added to the discussion.
Author Response
#Response to Reviewer 1
- Was the advised caloric intake of patients calculated or measured, when providing initial advice to patients? In the absence of indirect calorimetry, nutritional counseling could have caused patients to take either too much or too few calories (due to formulas often missing the actual correct intake), which should be added to the discussion.
Response: Thank you for providing this guidance. In order to better align with clinical practice, this study estimated the daily total energy requirements of patients by utilizing the simplified calorie algorithm recommended by the Ministry of Health and Welfare in Taiwan. This estimation takes into account the body mass index of the subjects (underweight, normal, overweight, or obese) and their daily activity levels (light, moderate, or heavy). For further details on the estimation process, please refer to the following link: https://www.hpa.gov.tw/Pages/Detail.aspx?nodeid=168&pid=728.
We fully endorse the point raised that without indirect calorimetry, nutritional counseling could result in patients consuming either excessive or insufficient calories (due to formulas often inaccurately determining the correct intake). Consequently, in the "Limitations" section of the revised manuscript within 4. Discussion (page 16, lines 315-320), we have included content addressing this issue.
We are grateful for the chance to revise our work and sincerely hope that our revisions meet your approval. We appreciate your valuable time and effort in guiding us to enhance the paper.
Reviewer 2 Report
Comments and Suggestions for Authors
The authors should be congratulated for their work. The study is very interesting and poses a focus on a topic that is often underestimated and should be more taken care of. Nutrition is part of daily lifestyle, introit and excretory functions are therefore deeply influenced by these kind of factors.
The methods that are used are clear and valid, with strong sounding results. More care should be applied to nutrition as a factor that may influence long term post-operative outcomes.
Author Response
Thank you for your comprehensive review. We appreciate your valuable time and effort.